# Genome-Wide Identification and Expression Analysis of Kiwifruit Leucine-Rich Repeat Receptor-Like Proteins Reveal Their Roles in Biotic and Abiotic Stress Responses

**DOI:** 10.3390/ijms25084497

**Published:** 2024-04-19

**Authors:** Yingying Cao, Congxiao Zhang, Fang Liu, Dawei Li, Aidi Zhang, Li Li, Xiujun Zhang

**Affiliations:** 1Key Laboratory of Plant Germplasm Enhancement and Specialty Agriculture, Wuhan Botanical Garden, Chinese Academy of Sciences, Wuhan 430074, China; caoyingying21@mails.ucas.ac.cn (Y.C.); zhangcongxiao21@mails.ucas.ac.cn (C.Z.); liufang@wbgcas.cn (F.L.); lidawei@wbgcas.cn (D.L.); zhangaidi@wbgcas.cn (A.Z.); 2Center of Economic Botany, Core Botanical Gardens, Chinese Academy of Sciences, Wuhan 430074, China; 3University of Chinese Academy of Sciences, Beijing 100049, China

**Keywords:** kiwifruit, *LRR-RLP*, stress, differentially expressed gene, plant immunity

## Abstract

Leucine-rich repeat receptor-like proteins (*LRR-RLPs*), a major group of receptor-like proteins in plants, have diverse functions in plant physiology, including growth, development, signal transduction, and stress responses. Despite their importance, the specific roles of kiwifruit *LRR-RLPs* in response to biotic and abiotic stresses remain poorly understood. In this study, we performed family identification, characterization, transcriptome data analysis, and differential gene expression analysis of kiwifruit *LRR-RLPs*. We identified totals of 101, 164, and 105 *LRR-RLPs* in *Actinidia chinensis* ‘Hongyang’, *Actinidia eriantha* ‘Huate’, and *Actinidia chinensis* ‘Red5’, respectively. Synteny analysis revealed that the expansion of kiwifruit *LRR-RLPs* was primarily attributed to segmental duplication events. Based on RNA-seq data from pathogen-infected kiwifruits, we identified specific *LRR-RLP* genes potentially involved in different stages of pathogen infection. Additionally, we observed the potential involvement of kiwifruit *LRR-RLPs* in abiotic stress responses, with upstream transcription factors possibly regulating their expression. Furthermore, protein interaction network analysis unveiled the participation of kiwifruit LRR-RLP in the regulatory network of abiotic stress responses. These findings highlight the crucial roles of *LRR-RLPs* in mediating both biotic and abiotic stress responses in kiwifruit, offering valuable insights for the breeding of stress-resistant kiwifruit varieties.

## 1. Introduction

Kiwifruit (*Actinidia* Lindl.), classified in the Actinidiaceae family and the genus *Actinidia*, is commonly referred to as the “King of Fruits” due to its exceptionally high contents of vitamin C, minerals, dietary fiber, and other beneficial metabolites [1,2,3]. Originating in China, kiwifruit has become a significant cash crop, gaining popularity worldwide for its flavor and nutritional attributes [1]. Given its high nutritional, medicinal and economic value, kiwifruit holds a promising future [2,3,4,5]. However, throughout the growth and development of kiwifruit, it is inevitably influenced by various biotic or abiotic stresses, such as infection by pathogenic bacteria, heat stress, waterlogging stress, and metal stress. Notably, the emergence of *Pseudomonas syringae* pv. *actinidiae* (Psa) disease has proven to be a serious global threat to kiwifruit, leading to substantial economic losses in the industry on a worldwide scale [6,7]. Heat stress poses another challenge, as it can adversely affect the growth and development of plants [8]. Kiwifruit roots, being shallow and sensitive to water, can experience reduced crop yields and even plant fatalities [9]. Therefore, it is essential to prioritize research on the resistance response mechanisms in kiwifruit. Investigating these mechanisms is crucial for developing strategies to ensure the economic viability and sustainability of kiwifruit cultivation in the face of diverse stress challenges.

Plants exist in a dynamically changing environment, where they must strike a balance between harnessing benefits from microorganisms and fending off pathogenic challenges. While beneficial microorganisms contribute to plant growth and agricultural production, pathogenic ones can result in diseases and economic losses. Unlike animals, plants lack a circulating immune system and depend on each cell’s ability to deploy innate immune responses against potential pathogens [10]. Plant immune receptors recognize the presence and activities of specific microorganisms, triggering coordinated downstream signaling pathways and initiating immune responses [11,12]. Pattern recognition receptors (PRRs) include receptor-like kinases (RLKs) and receptor-like proteins (RLPs) [10,13]. RLKs consist of an extracellular domain for ligand recognition, a single-pass transmembrane domain for intracellular and extracellular signaling and membrane localization. Additionally, they feature an intracellular kinase domain responsible for transmitting immune signals. In contrast, RLPs carry a short cytoplasmic tail and lack the intracellular kinase domain [14,15]. Due to the intracellular kinase domain, RLKs can independently trigger signals, whereas RLPs need to form a functional complex with RLKs to activate downstream signals [16]. According to the different extracellular domains, PRRs can be categorized into LRR domain receptors, LysM domain receptors, Lectin domain receptors, and EGF-like domain receptors [17,18]. Within the RLPs family, *LRR-RLPs* constitute the largest subfamily [19].

*LRR-RLP* plays a crucial role in plant growth and development, signal transduction and stress responses. For instance, CLAVAT2 (CLV2)/AtRLP10 regulate shoot apical meristem maintenance and differentiation, impacting organ development [20]. In *Arabidopsis thaliana*, *CLV2* is essential for meristem development, and its maize homolog *FASCINATED EAR2* (*FEA2*) also governs meristem development [21]. Another example is *TOO MANY MOUTHS* (*TMM*)/*AtRLP17* in *A. thaliana*, which contributes to its overall development and plays a key role in regulating stomatal distribution on the epidermis [22,23]. *AtRLP41* modulates leaf sensitivity to abscisic acid (ABA) and potentially participates in ABA-induced senescence [23]. AtRLP44 activates BR signaling by directly interacting with the BR coreceptor BAK1 [24]. In tomato, the *Cf-9* gene is vital for pathogen defense and confers resistance to *Avr9* [25]. The expression of OsRLP1 significantly increases after infection with rice black-streaked dwarf virus (RBSDV), and its interaction with the receptor-like kinase OsSOBIR1 induces PTI response and enhances antiviral defense [26]. Additionally, *LRR-RLPs* play a crucial role in abiotic stress tolerance. For instance, *AtRLP28* overexpressed plants exhibit enhanced salt stress tolerance [27].

The *LRR-RLP* gene family has undergone extensive study across various plant species. In the model plant *A. thaliana*, the *LRR-RLP* gene family was identified, revealing 57 *LRR-RLP* members [23]. A genome-wide analysis of wild bananas identified 78 members in the *LRR-RLP* family [28]. Poplar has 82 identified *LRR-RLPs* [29]. In tomato, a total of 176 *LRR-RLP* members were identified, nearly three times the number in *Arabidopsis* [21]. Rice has 90 *LRR-RLPs* [30]. *Brassica napus*, *Brassica juncea*, *Brassica rapa*, and *Brassica nigra* possess 276, 226, 63, and 175 *LRR-RLPs* members, respectively [19,31].

Kiwifruit is vulnerable to both biotic and abiotic stresses. *LRR-RLP* is crucial for plant growth and development, signal transduction, and stress responses. However, its study in kiwifruit has received limited attention. This study delves into the analysis of the *LRR-RLP* genes in kiwifruit, encompassing the identification of gene family members, phylogenetic analysis, chromosomal localization, synteny analysis, transcriptome data analysis, upstream transcription factor analysis, and construction of a protein interaction network. Our findings contribute to a deeper understanding of the defense mechanisms employed by kiwifruit, holding crucial reference value for the genetic enhancement of kiwifruit varieties. Moreover, this study provides valuable insights for the development of resistant kiwifruit varieties.

## 2. Results

### 2.1. Identification of LRR-RLP Gene Family in Kiwifruit

To determine the number of *LRR-RLP* family members in ‘Hongyang’, ‘Huate’, and ‘Red5’ kiwifruit, we retrieved 57 LRR-RLP sequences from *Arabidopsis* databases. BLASTP searches were conducted using these sequences as queries in three kiwifruit databases, with an e-value threshold set at 1 × 10^−5^. The obtained candidate sequences were then submitted to the Pfam database to confirm the presence of the LRR domain in each candidate sequence. The results revealed the following counts: 101 *LRR-RLPs* in ‘Hongyang’ kiwifruit (*HyLRR-RLPs*), 164 *LRR-RLPs* in ‘Huate’ kiwifruit (*HtLRR-RLPs*), and 105 *LRR-RLPs* in ‘Red5’ kiwifruit (*DhLRR-RLPs*).

We also conducted an analysis of the physicochemical properties of LRR-RLP members in these three kiwifruit species (Figure 1). Detailed physicochemical properties of HyLRR-RLPs, HtLRR-RLPs, and DhLRR-RLPs are provided in Appendix A, respectively. The amino acid sequence lengths of HyLRR-RLPs ranged from 117 to 2136, with their protein molecular weights (MWs) ranging from 12.60 to 234.25 kDa, and isoelectric points (PIs) ranging between 4.45 and 10.3. For HtLRR-RLPs, the amino acid sequence lengths ranged from 74 to 1260, the protein MWs were from 8.29 to 138.89 kDa, and the PIs ranged between 4.3 and 12.11. The amino acid sequences of DhLRR-RLPs ranged from 77 to 1180, the protein MWs varied from 8.90 to 129.82 kDa, and the PIs were between 4.19 and 9.51.

### 2.2. Phylogenetic Relationships of the Kiwifruit LRR-RLP Gene Family Members

To understand the evolutionary relationship between ‘Hongyang’ and ‘Huate’ kiwifruit, we performed multiple sequence alignment using MAFFT v7.490 for 101 HyLRR-RLPs and 164 HtLRR-RLPs identified in this study. Subsequently, a phylogenetic tree was constructed using the maximum likelihood method with IQtree v2.0.3 for Linux 64-bit (Figure 2). Based on the clustering results of the phylogenetic tree, we divided it into 9 groups. The distributions of ‘Hongyang’ kiwifruit and ‘Huate’ kiwifruit within each group are listed in Table 1. Notably, group IX exhibited the highest count with 71 members, including 25 HyLRR-RLPs and 46 HtLRR-RLPs. Group VIII followed closely, with 57 members, consisting of 22 HyLRR-RLPs and 35 HtLRR-RLPs. Conversely, group IV housed the fewest LRR-RLPs members, with 1 HyLRR-RLPs and 2 HyLRR-RLPs. The phylogenetic tree in Figure 2 and Table 1 illustrates a discernible clustering pattern, indicating a trend of convergent evolution of the kiwifruit LRR-RLP gene.

### 2.3. Chromosomal Localization Analysis of Kiwifruit LRR-RLP Gene Family Members

We analyzed the chromosomal distributions of the *HyLRR-RLPs*, *HtLRR-RLPs*, and *DhLRR-RLPs* identified in this study (Appendix A). The results revealed that among the 101 *HyLRR-RLP* family members, 88 were situated in 28 linked groups, while the remaining 13 were found in 13 unlinked groups. Notably, LG07 of ‘Hongyang’ kiwifruit did not contain any *LRR-RLP* genes. Regarding the *HtLRR-RLPs*, all 164 genes were dispersed among 29 chromosomes. For the *DhLRR-RLPs*, 100 genes were dispersed among 29 linked groups. The distribution pattern of kiwifruit *LRR-RLPs* appears to be extensive, with genes dispersed across linked groups and chromosomes.

### 2.4. Synteny Analysis of LRR-RLPs in Kiwifruit

We performed synteny analysis to reveal duplication events and possible collinear blocks within or between kiwifruit genomes. Collinearity analysis of the LRR-RLPs between the genomes of kiwifruit and *Arabidopsis* revealed that 6 HyLRR-RLPs exhibited syntenic relationships with 4 AtLRR-RLPs, while 3 HtLRR-RLPs exhibited syntenic relationships with 4 AtLRR-RLPs. Additionally, 9 DhLRR-RLPs exhibited syntenic relationships with 8 AtLRR-RLPs (Appendix A). Synteny analysis between kiwifruit genomes showed that 35 HyLRR-RLPs and 34 HtLRR-RLPs formed 45 pairs, 1 HyLRR-RLPs and 1 DhLRR-RLP formed 1 pair, and 43 HtLRR-RLPs and 52 DhLRR-RLPs formed 65 pairs (Appendix A). What is more, we carried out synteny analysis within the kiwifruit. The results revealed that HyLRR-RLPs exhibited 46 pairs of segmental duplication and 3 pairs of tandem duplication, HtLRR-RLPs exhibited 56 pairs of segmental duplication and 9 pairs of tandem duplication, and DhLRR-RLPs exhibited 51 pairs of segmental duplication and 7 pairs of tandem duplication (Appendix A).

The Ka/Ks values were calculated to evaluate the selection pressure during evolution [3]. We found that the Ka/Ks values of the duplicate pairs identified above were mostly less than 1, indicating that these genes evolved under purifying selection.

### 2.5. Expression Profiles of LRR-RLPs in Response to Psa Infection

Based on the transcriptome data of kiwifruit infected with Psa, we analyzed the expressions of *LRR-RLP* genes (Figure 3). Our analysis revealed differentially expressed *LRR-RLP* genes in both resistant ‘Huate’ kiwifruit (Ht) and susceptible ‘Hongyang’ kiwifruit (Hy) during Psa infection. Specifically, 10 *LRR-RLPs* genes were differentially expressed in Hy, and 22 *LRR-RLP* genes were differentially expressed in Ht. We found that the number of *LRR-RLPs* in Ht was more than that in Hy, regardless of the differentially up-regulated expression or differentially down-regulated expression (Figure 4a,b).

In addition, with the increase in infection time, the differentially up-regulated LRR-RLP genes in Ht showed a gradual increase, and the differentially down-regulated LRR-RLP genes showed a gradual decrease, but there was no such change in Hy. Compared to the 0 h time point, we analyzed the differentially expressed *LRR-RLP* genes in Hy and Ht at 12 h, 24 h, 48 h, and 96 h, respectively (Figure 4c). Simultaneously, we detected the differentially expressed *LRR-RLP* genes between the two kiwifruit types at the same time point (Figure 4d). In Hy, we identified 10 differentially expressed *LRR-RLP* genes, among which 5, 6, 8, and 5 genes were differentially expressed at 12 h, 24 h, 48 h, and 96 h, respectively. In Ht, we found 22 differentially expressed *LRR-RLP* genes, with 9, 9, 12, and 14 differentially expressed genes at 12 h, 24 h, 48 h, and 96 h, respectively. Of these genes, 4 are unique to Hy and 16 are unique to Ht. Compared to the 0 h time point, the numbers of *LRR-RLP* genes that were simultaneously differentially expressed in both Hy and Ht at the 12 h, 24 h, 48 h, and 96 h time points were 2, 2, 2, and 2, and these two genes were *Actinidia39875.t1* and *Actinidia06882.t1*. Using Hy as a control, the comparison of the differentially expressed *LRR-RLP* genes between Hy and Ht at the same time points revealed 14, 21, 16, 19, and 14 genes at 0 h, 12 h, 24 h, 48 h, and 96 h, respectively (Figure 4d).

In Figure 5, several genes exhibit interesting expression changes. *Actinidia39875.t1* was differentially expressed in both Hy and Ht, with a higher expression in Ht kiwifruit at each time point. In Hy, the expression of *Actinidia39875.t1* gradually increased, while in Ht, it initially increased, peaked at 12 h, and then gradually decreased. *Actinidia35026.t1* was not differently expressed in Hy, but was differently expressed at 48 h and 96 h in Ht. The expression of *Actinidia35026.t1* remained stable in ‘Hongyang’ kiwifruit and gradually decreased in ‘Huate’ kiwifruit. *Actinidia12020.t1* and *Actinidia34764.t1* in Hy had relatively constant expression levels with no differential expression. However, their expression levels show a gradual increasing trend in Ht, and both were differentially up-regulated at the 48 h and 96 h time points. This suggests that these two genes in Ht consistently respond to Psa infection. According to these results, we speculate that the four genes *Actinidia39875.t1*, *Actinidia35026.t1*, *Actinidia12020.t1*, and *Actinidia34764.t1* play a role in the process of Psa infection. *Actinidia39875.t1*, and *Actinidia35026.t1* play a role in the early stage of Psa infection, while *Actinidia12020.t1* and *Actinidia34764.t1* play a role in the late stage of Psa infection.

### 2.6. Analysis of Upstream Transcription Factors of LRR-RLP in Kiwifruit

To determine the potential pathways governing the expressions of *LRR-RLP* genes in kiwifruit, we intersected the genes differentially expressed in both Hy and Ht and subsequently identified the upstream transcription factors for these genes using the PlantRegMap database. After filtering out the transcription factors with low expression and no significant changes, as well as those with no differential expression in Hy and Ht, we obtained 40 transcription factors (Figure 6). Compared to the 0 h time point, 9, 15, 15, and 8 transcription factors displayed differential expressions in Hy at 12 h, 24 h, 48 h, and 96 h, respectively. In Ht, 24, 20, 16, and 15 transcription factors were differentially expressed at 12 h, 24 h, 48 h, and 96 h, respectively. Among these, 7 were unique to Hy, while 15 were unique to Ht (Table 2). Notably, the number of unique differentially expressed transcription factors in Ht was approximately twice that in Hy.

In order to intuitively observe the intricate relationship between the 40 transcription factors and the 25 genes, we plotted a network diagram, as illustrated in Figure 7. Utilizing the PlantRegMap database, we further obtained the families of these 40 transcription factors (Appendix A). Among them, 15 transcription factors exclusively differentially expressed in Ht belong to the MYB, TCP, WRKY, TALE, HD-ZIP, ERF, bZIP, and C2H2 families. Specially, *Actinidia29458.t1* serves as an upstream transcription factor for *Actinidia36397.t1* and *Actinidia21924.t1*. *Actinidia14109.t1* is an upstream transcription factor for *Actinidia17147.t1* and *Actinidia21924.t1*. Both *Actinidia29458.t1* and *Actinidia14109.t1* belong to the ERF family, and this family is involved in regulating plant development and tolerance to biotic/abiotic stresses [32]. *Actinidia10240.t1*, a member of the TALE family, functions as an upstream transcription factor for *Actinidia02532.t1*, *Actinidia00252.t1*, *Actinidia09221.t1*, *Actinidia10997.t1*, *Actinidia12822.t1*, *Actinidia36397.t1*, *Actinidia39532.t1*, *Actinidia21924.t1*, and *Actinidia12020.t1*. The TALE family is associated with meristem formation and the regulation of leaf morphology [33]. Furthermore, *Actinidia37473.t1* is an upstream transcription factor of *Actinidia14325.t1*. *Actinidia39836.t2* is an upstream transcription factor for *Actinidia14325.t1*, *Actinidia17147.t1*, and *Actinidia34764.t1*. *Actinidia37473.t1* and *Actinidia39836.t2* belong to the C2H2 gene family, which is involved in plant development, and can control chloroplast development and starch granule formation [34]. These differentially expressed transcription factors have the potential to regulate the expression of kiwifruit *LRR-RLP* genes, thereby influencing plant resistance.

### 2.7. Expression Profiles of LRR-RLPs in Response to Abiotic Stresses

To explore the impact of kiwifruit *LRR-RLPs* under abiotic stress conditions, we conducted an analysis using transcriptome data from kiwifruit subjected to three different abiotic stresses: heat, waterlogging, and copper. The results reveal distinct responses of *LRR-RLP* genes in kiwifruit under these different stresses (Figure 8). Specially, 31 *LRR-RLP* genes were differentially expressed under waterlogging stress, 27 under heat stress, and 20 under copper stress. Notably, the genes *Acc09317*, *Acc13131*, *Acc16306*, *Acc17095*, *Acc18550*, and *Acc32474* were all responsive to waterlogging stress, heat stress, and copper stress. This suggests that *LRR-RLPs* may be involved in the response to multiple abiotic stresses, highlighting their potential key regulatory role in enhancing kiwifruit resistance to adverse environmental conditions.

### 2.8. Protein Interaction Network

Taking the control group as a reference, we compared the expression data under three abiotic stresses, i.e., heat, copper, and waterlogging, and screened the differentially expressed genes in each case. The identified differentially expressed genes were then overlapped. Subsequently, we constructed an interaction network between the LRR-RLPs and the proteins encoded by these differentially expressed genes in kiwifruit (Figure 9). This protein interaction network contains 341 proteins, among which 9 are LRR-RLP proteins, i.e., Acc03124, Acc02009, Acc13584, Acc16306, Acc27351, Acc30038, Acc01049, Acc03001, and Acc17095. Notably, interactions were observed between the LRR-RLPs and 9 other proteins in kiwifruit, i.e., Acc17013, Acc23455, Acc19741, Acc04947, Acc17267, Acc29190, Acc06282, Acc10774, and Acc29538. Through interactions with these proteins, kiwifruit LRR-RLPs regulate downstream proteins within the network. These *LRR-RLP* members may play a crucial role in regulating the abiotic stress tolerance of kiwifruit.

## 3. Discussion

In this study, we performed identification, classification, chromosomal localization, and synteny analysis of *LRR-RLPs* in three kiwifruit species. Totals of 101, 164, and 105 *LRR-RLPs* were identified in ‘Hongyang’, ‘Huate’ and ‘Red5’ kiwifruit, respectively. HyLRR-RLPs and HtLRR-RLPs were classified into nine groups, with the highest and lowest numbers observed in group IX and group IV, respectively. The LRR-RLPs of the three species of kiwifruit were widely distributed and dispersed in their chromosomes, linked groups, or unlinked groups. Specifically, HyLRR-RLPs, HtLRR-RLPs, and DhLRR-RLPs exhibited the highest abundance on LG07, Chr16, and LG01, respectively, with counts of 7, 17, and 9. Collinearity analysis revealed 6 pairs of collinear relationships between AtLRR-RLP and HyLRR-RLPs, involving 10 genes (4 AtLRR-RLPs and 6 HyLRR-RLPs). HtLRR-RLPs exhibited four pairs of collinear relationships with AtLRR-RLPs, involving seven genes (four AtLRR-RLPs and three HtLRR-RLPs). AtLRR-RLPs and DhLRR-RLPs displayed 11 collinearity relationships, involving 17 genes (8 AtLRR-RLPs and 9 DhLRR-RLPs). There were 45 covariate pairs between HyLRR-RLPs and HtLRR-RLPs involving 69 genes. Only one collinear relationship involving two genes was observed between HyLRR-RLP and DhLRR-RLP. Meanwhile, HtLRR-RLPs and DhLRR-RLPs exhibited 65 pairs of collinearity, involving 95 genes. There were 46, 56, and 51 pairs of segmental duplications in the Hy, Ht, and Dh genomes, respectively. Additionally, there were 3, 9, and 7 pairs of tandem duplications in the genomes of Hy, Ht, and Dh, respectively. The results suggest that the amplification of the *LRR-RLP* gene family in kiwifruit was mainly due to segmental duplication. Furthermore, the Ka/Ks values for the identified duplicate pairs were mostly less than 1, indicating that these genes evolved under purifying selection.

Kiwifruit is an important economic crop. However, it faces challenges in yield and quality caused by adverse environmental conditions during growth and development. *LRR-RLPs* encounter various external and internal cellular signals during the growth and development stages, triggering signal transduction pathways and biological responses. As a pattern recognition receptor, *LRR-RLPs* play an important role in plant growth and development, signal transduction, and stress response. Both *Arabidopsis CLV2* and its functional homologous protein *FEA2* in maize regulate the maintenance and differentiation of meristem and the development of related organs [20,21]. *AtRLP41* and *AtRLP44* are involved in the signal transduction of plant hormones [23,24]. RBPG1/AtRLP42 serves as a novel microbial-associated molecular pattern receptor, recognizing fungal endopolygalacturonases (PGs) [35]. Several LRR-RLP proteins have been implicated in plant disease resistance, including the apple HcrVf2 protein, the Cf protein of tomato, and the Ve protein of tomato [21,23]. However, the functions of LRR-RLPs and their response to stress in kiwifruit remain to be elucidated.

Using RNA-seq data obtained from both resistant and susceptible kiwifruits infected with Psa, we analyzed the differentially expressed *LRR-RLP* genes at various time points. Furthermore, we analyzed the differentially expressed *LRR-RLP* genes in both kiwifruit species at different time points compared to the 0 h time point. Notably, we observed a higher number of differentially expressed *LRR-RLP* genes in Ht compared to Hy, with the genes unique to Ht being four times more abundant than those in Hy. Furthermore, we analyzed the differentially expressed *LRR-RLP* genes between the two kiwifruit species at the same time points. Interestingly, some genes exhibited insignificant changes in expression in Hy but showed significant trends in Ht, specifically *Actinidia39875.t1*, *Actinidia35026.t1*, *Actinidia12020.t1*, and *Actinidia34764.t1*. These findings suggest a pivotal role of LRR-RLP in the process of Psa infection in kiwifruit. Moreover, our analysis extended to abiotic stresses, revealing that 31, 27, and 20 *LRR-RLP* genes were differentially expressed under waterlogging, heat, and copper stress, respectively. This indicates a crucial role of *LRR-RLPs* in enhancing the resistance of kiwifruit to abiotic stresses. The comprehensive examination of both biotic and abiotic stress responses highlights the multifaceted involvement of *LRR-RLPs* in shaping the defense mechanisms of kiwifruit. Most of the upstream transcription factors of the differentially expressed *LRR-RLP* genes in Hy and Ht belong to families associated with plant growth, development, and stress response. In the interaction network involved in kiwifruit LRR-RLPs, there were 341 proteins, including 9 LRR-RLP proteins. Homologous genes coding for these proteins were searched and annotated in the *Arabidopsis* TAIR and Uniprot databases. For instance, the homologous gene *SERK2* of *Acc04947* played an important role in zygotic embryo development [36]. The *Acc17267* homolog *ATLOX1* was highly expressed in *Arabidopsis* roots and seedlings under the induction of pathogens and plant hormones and involved in lateral root development and defense response [37]. These findings suggest potential functional implications for the identified homologous genes in kiwifruit LRR-RLP associated processes. We speculate that LRR-RLPs and their related upstream transcription factors play an important role in kiwifruit response to biotic and abiotic stresses.

## 4. Materials and Methods

### 4.1. Data Resource

The genome files and gff3 files of ‘Hongyang’, ‘Red5’, and ‘Huate’ kiwifruit were sourced from the kiwifruit genome database (http://kiwifruitgenome.org/, accessed on 16 September 2022) [38]. Fifty-seven *Arabidopsis LRR-RLP* (AtLRR-RLP) sequences were retrieved from the *Arabidopsis* database (https://www.arabidopsis.org/, accessed on 8 September 2023). Transcriptome data were obtained from NCBI database (http://www.ncbi.nlm.nih.gov/, accessed on 20 September 2023). The transcriptome data registration numbers are as follows: *Pseudomonas syringae* pv. *actinidiae* infected kiwifruit: PRJNA514180; kiwifruit under heat stress: PRJNA796069; kiwifruit under waterlogging stress: PRJNA765913.

### 4.2. Genome-Wide Identification of LRR-RLP Genes in Kiwifruit

The 57 AtLRR-RLPs obtained were subjected to a BLASTP search against the protein databases of ‘Hongyang’, ‘Huate’, and ‘Red5’ kiwifruit, respectively. The e-value threshold was set to 1 × 10^−5^ to reduce false positives. The putative ‘Hongyang’ LRR-RLPs (HyLRR-RLPs), ‘Huate’ LRR-RLPs (HtLRR-RLPs), and ‘Red5’ LRR-RLPs (DhLRR-RLPs) were then submitted to the Pfam database to identify their protein sequences containing LRR domains [39]. Key features, such as the amino acid length, isoelectric point, and molecular weight of HyLRR-RLP, HtLRR-RLP, and DhLRR-RLP proteins, were analyzed using the ProtParam online tool (http://au.expasy.org/tools/protparam.html, accessed on 30 October 2023) [40].

### 4.3. Multiple Sequence Alignment and Phylogenetic Tree Analysis

We used MAFFT v7.490 to perform the multiple sequence alignment of the HyLRR-RLPs and HtLRR-RLPs, with the parameters set to their default [41]. Subsequently, we employed IQtree software v2.0.3 for Linux 64-bit to construct a phylogenetic tree for both the HyLRR-RLPs and HtLRR-RLPs using the maximum likelihood method [42]. The resulting tree was further visualized and modified by Figtree v1.4.4 (http://tree.bio.ed.ac.uk/software/figtree/, accessed on 30 October 2023).

### 4.4. Chromosomal Localization and Collinearity Analysis

The chromosome coordinates of the HyLRR-RLPs, HtLRR-RLPs, and DhLRR-RLPs were extracted from the genome files and GFF3 annotation files of ‘Hongyang’, ‘Huate’, and ‘Red5’ kiwifruit. To comprehend their distribution on the chromosomes, we generated distribution maps using the MG2C v2.1 online tool (http://mg2c.iask.in/mg2c_v2.1/, accessed on 14October 2023) [43]. 

For collinearity analysis, we employed the one-step MCScanX-super Fast plugin in TBtools v2.06 to investigate the collinearity between *Arabidopsis* and the three kiwifruit species, as well as among the three kiwifruit species, with an E-value threshold of 0.001 [44]. Additionally, block analysis of the three kiwifruit species was carried out using the same method.

### 4.5. Transcriptome Data Analysis

To investigate the expression patterns of kiwifruit *LRR-RLPs* in response to biotic and abiotic stresses, we conducted a comprehensive analysis of multiple transcriptome datasets. For the examination of *LRR-RLP* expression in kiwifruit responding to Psa infection, we performed a detailed analysis. The Psa infection time points were 0 h, 12 h, 24 h, 48 h, and 96 h, with three biological replicates for each time point. The heat stress treatment included 0 h, 2 h, and 4 h at 50 degrees Celsius, with three biological replicates for each time node. Waterlogging stress was applied over 0, 1, and 2 days, each with three biological replicates.

We performed the subsequent processing steps on the provided transcriptome data. Firstly, we assessed the quality of the transcriptome data with the FastQC tool v0.11.9 [45], and then applied fastp v0.11.4 to obtain clean reads [46]. Secondly, we aligned the clean reads from all sample data with the kiwifruit reference genome using HISAT2 v2.2.1, resulting in SAM files [47]. Thirdly, we employed SAMtools v1.9 to convert the SAM files obtained earlier into BAM files, followed by sorting [48,49]. Lastly, we used Stringtie v2.2.1 for further transcriptome assembly and read sequence quantification [50]. The gene expression levels were measured by FPKM (transcribed fragments per kilobase per million of labeled reads). Differential gene expression analysis was performed using EdgeR [51]. The threshold to identify differentially expressed genes was *p*-value < 0.05 and logFC ≥ 1. Subsequently, based on the *LRR-RLP* gene expression matrix, cluster analysis was performed by the HeatMap plug-in in TBtools v2.06 [44].

### 4.6. Analysis of Upstream Transcription Factors of LRR-RLPs in Kiwifruit

We compiled a concatenated set of genes that exhibited differential expression in ‘Hongyang’ and ‘Huate’. Subsequently, we identified the upstream transcription factors associated with these genes using the PlantRegMap database (http://plantregmap.gao-lab.org/, accessed on 8 January 2024) [33]. This tool facilitates the inference of potential interactions between the input genes and transcription factors. Finally, we visualized the potential regulatory relationships by Cytoscape v3.8.2 [52].

### 4.7. Construction of a Protein Interaction Network

Taking the control group as a reference, we performed a comparative analysis of the expression data under three abiotic stresses, i.e., heat, copper, and waterlogging. Differentially expressed genes were identified through screening, and the overlapping set of these genes was determined. In the STRING database (https://cn.string-db.org/, accessed on 13 January 2024) [53], the protein sequences of kiwifruit LRR-RLPs and the previously identified differentially expressed genes were used as input files. *A. thaliana* was chosen as the reference species for the comparison of the protein sequences. The protein sequences of the homologous genes were subsequently mapped to their interactions in the database, leading to the formation of a comprehensive interaction network. The parameters were set to a 70% reliability threshold and a 5% error detection rate.

## 5. Conclusions

In conclusion, this study identified and analyzed the kiwifruit LRR-RLP gene family members, their physicochemical properties, phylogenetic relationships, and chromosomal positions, and determined their expression levels under biotic and abiotic stress. Segmental duplication emerged as the primary driver for the expansion of the *LRR-RLP* gene family in kiwifruit. The differentially expressed *LRR-RLP* genes in resistant and susceptible kiwifruits infected with Psa highlight their involvement in biotic stress responses. We identified four LRR-RLP genes and related transcription factors that may be involved in pathogen stress. Additionally, the expression patterns of the *LRR-RLP* genes under the three abiotic stresses underscore their significance in abiotic stress tolerance. The protein interaction network elucidates the regulatory relationships among the *LRR-RLP* members and their associated proteins, shedding light on the complex network governing abiotic stress responses in kiwifruit. These findings provide valuable reference information for understanding protein interactions within the *LRR-RLP* gene family and contribute to the study of resistant kiwifruit varieties.

## Figures and Tables

**Figure 1 ijms-25-04497-f001:**
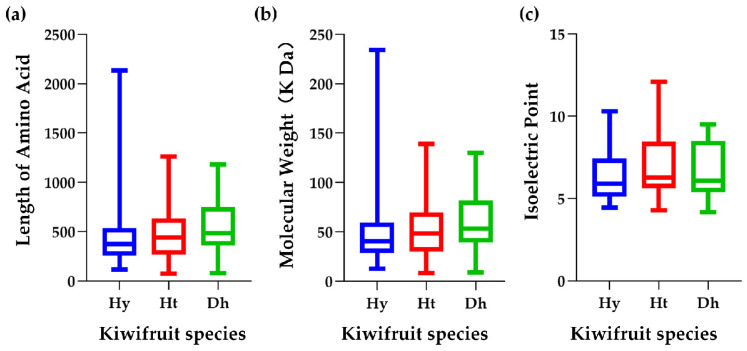
Physicochemical properties of LRR-RLPs in kiwifruit. (**a**) Length of amino acid in kiwifruit LRR-RLPs. (**b**) Molecular weight of kiwifruit LRR-RLPs. (**c**) Isoelectric point of kiwifruit LRR-RLPs. Different colors represent distinct kiwifruit species. Blue, red, and green represent ‘Hongyang’, ‘Huate’, and ‘Red5’ kiwifruit, respectively.

**Figure 2 ijms-25-04497-f002:**
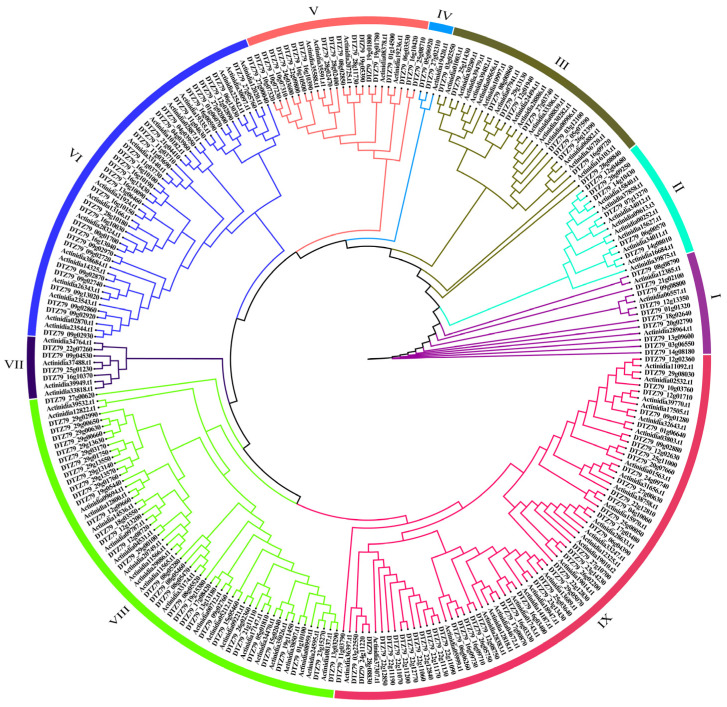
A phylogenetic tree of LRR-RLP members in ‘Hongyang’ and ‘Huate’ kiwifruit. The phylogenetic tree, derived from multiple sequence alignment with the maximum likelihood (ML) method, illustrates the relationships among the LRR-RLP members in ‘Hongyang’ and ‘Huate’ kiwifruit. The tree is categorized into 9 groups, with distinct colors representing the different phylogenetic groups.

**Figure 3 ijms-25-04497-f003:**
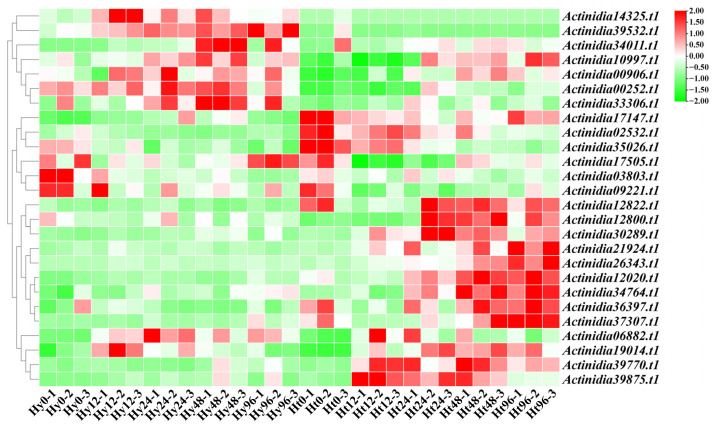
Expressions of *LRR-RLPs* after Psa infection in susceptible (Hy) and resistant (Ht) kiwifruit. Heatmaps of differentially expressed *LRR-RLPs* in susceptible and resistant kiwifruit after Psa infection. The X-axis represents the hours after Psa infection (0 h, 12 h, 24 h, 48 h, and 96 h), and the Y-axis represents the *LRR-RLP* genes.

**Figure 4 ijms-25-04497-f004:**
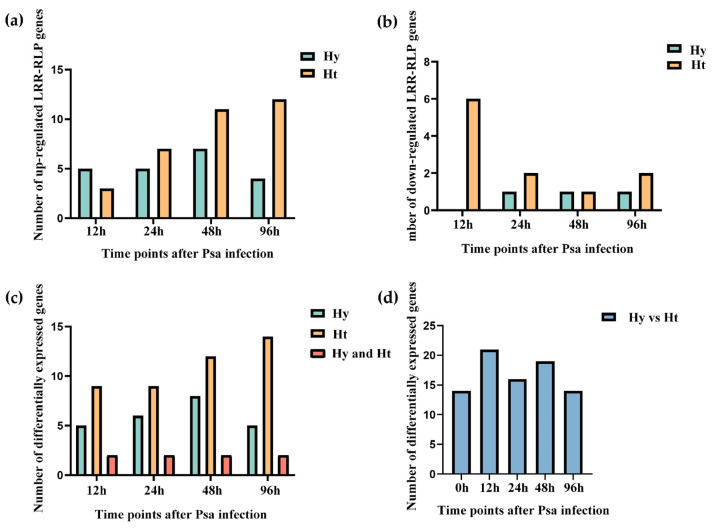
Differentially expressed *LRR-RLPs* after Psa infection. (**a**) Number of up-regulated *LRR-RLP* genes after infection by Psa in two kiwifruit species. (**b**) Number of down-regulated *LRR-RLP* genes after infection by Psa in two kiwifruit species. (**c**) Numbers of differentially expressed *LRR-RLPs* after Psa infection in two kiwifruit species. (**d**) Numbers of differentially expressed *LRR-RLPs* between the two kiwifruit types at the same time point.

**Figure 5 ijms-25-04497-f005:**
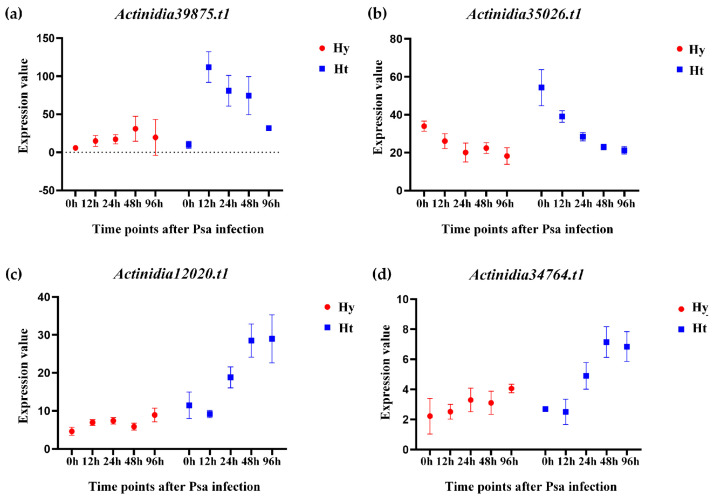
Expression levels of representative differentially expressed *LRR-RLPs* between susceptible (Hy) and resistant (Ht) kiwifruit after Psa infection. The X-axis represents the time point after Psa infection, and the Y-axis represents the gene expression value. (**a**) The expression value of *Actinidia39875.t1*. (**b**) The expression value of *Actinidia35026.t1*. (**c**) The expression value of *Actinidia12020.t1*. (**d**) The expression value of *Actinidia34764.t1*.

**Figure 6 ijms-25-04497-f006:**
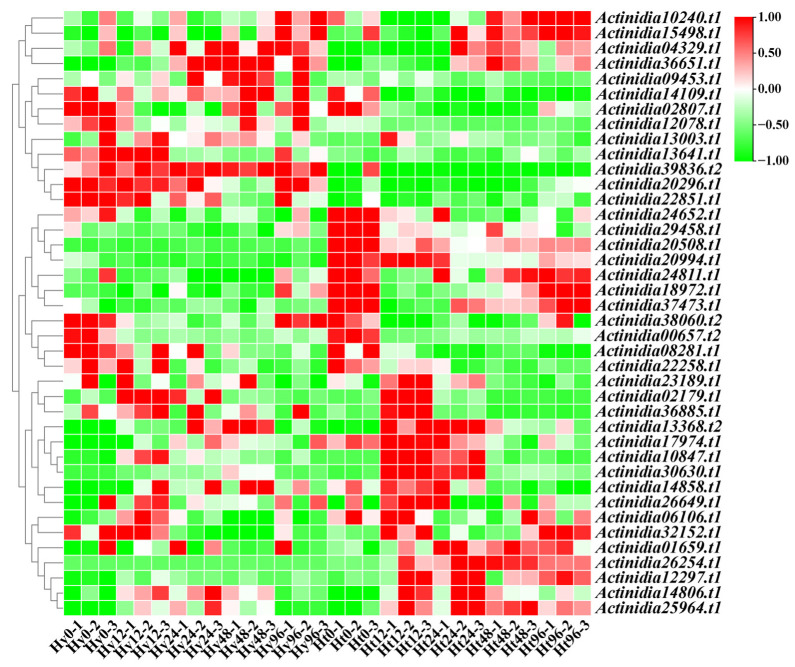
Heatmap of differentially expressed upstream transcription factors associated with differentially expressed LRR-RLP genes in two kiwifruit species.

**Figure 7 ijms-25-04497-f007:**
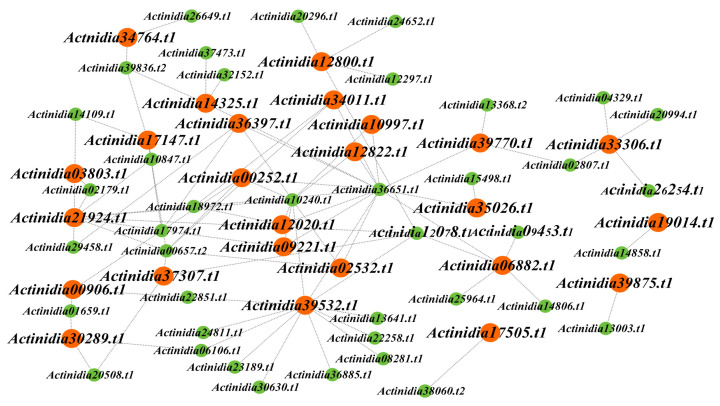
Regulatory network of *LRR-RLPs* and their corresponding transcription factors. The highlighted genes in this network are those that exhibit differential expression in at least one kiwifruit species. *LRR-RLP* genes are represented in orange, while transcription factors are denoted in green.

**Figure 8 ijms-25-04497-f008:**
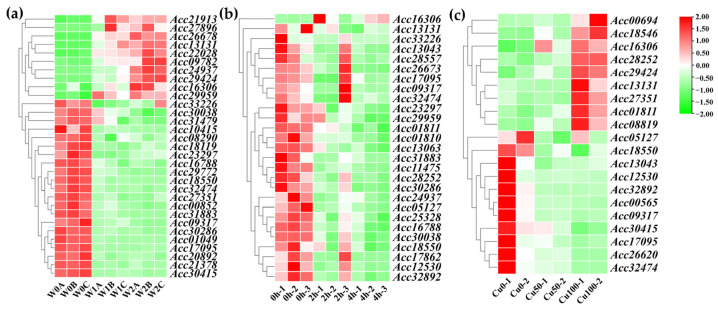
The expression profiles of differentially expressed *LRR-RLP* genes in kiwifruit waterlogging, heat, and copper stress. (**a**) Heatmap depicting differentially expressed kiwifruit *LRR-RLPs* under waterlogging stress. (**b**) Heatmap illustrating differentially expressed kiwifruit *LRR-RLPs* under heat stress. (**c**) Heatmap presenting differentially expressed kiwifruit *LRR-RLPs* under copper stress.

**Figure 9 ijms-25-04497-f009:**
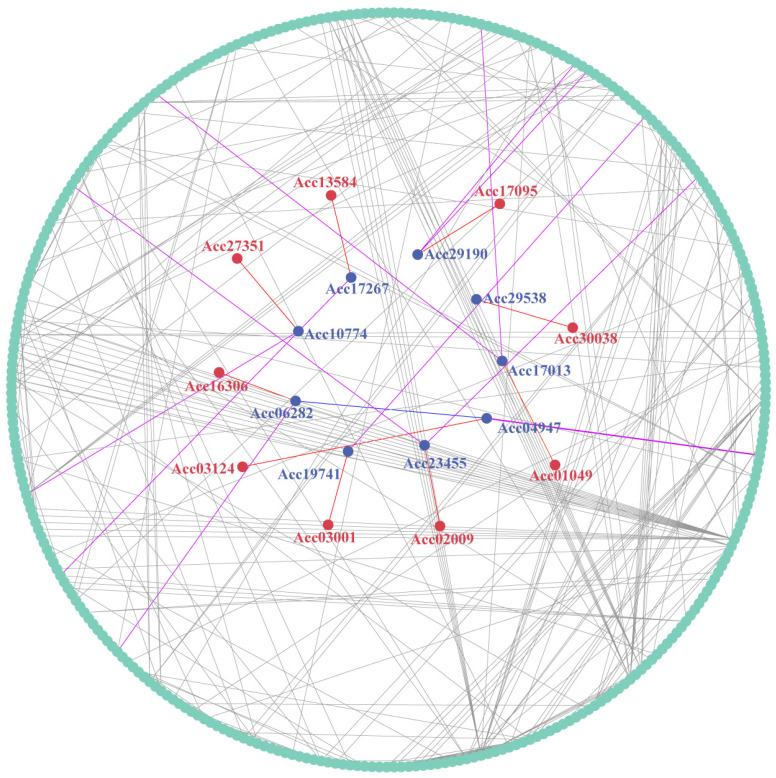
Protein interaction network involving kiwifruit LRR-RLPs. Red circles represent the LRR-RLP proteins, while blue circles represent other proteins associated with LRR-RLPs. The red lines connect the LRR-RLP proteins to other associated proteins, purple lines connect proteins associated with LRR-RLPs and other proteins associated with them, and blue lines link to other proteins associated with LRR-RLPs. The gray lines connect proteins encoded by genes that are differentially expressed, except for LRR-RLPs and other proteins associated with them.

**Table 1 ijms-25-04497-t001:** Numbers of *LRR-RLPs* in kiwifruit in different groups.

Groups	Number of HyLRR-RLPs	Number of HtLRR-RLPs	Total
I	3	10	13
II	9	6	15
III	16	11	27
IV	1	2	3
V	5	18	23
VI	16	32	48
VII	4	4	8
VIII	22	35	57
IX	25	46	71
Total	101	164	265

Notes: HyLRR-RLPs represent the LRR-RLPs of *Actinidia chinensis* ‘Hongyang’, HtLRR-RLPs represent the LRR-RLPs of *Actinidia eriantha* ‘Huate’, and DhLRR-RLPs represent the LRR-RLPs of *Actinidia chinensis* ‘Red5’.

**Table 2 ijms-25-04497-t002:** Transcription factors exclusively differentially expressed in one kiwifruit species.

Species	Exclusively Differentially Expressed Transcription Factors (Gene ID)
‘Hongyang’ (Hy)	*Actinidia17974.t1*, *Actinidia13641.t1*, *Actinidia32152.t1*,
	*Actinidia20296.t1*, *Actinidia14858.t1*, *Actinidia22851.t1*,
	*Actinidia15498.t1*
‘Huate’ (Ht)	*Actinidia01659.t1*, *Actinidia04329.t1*, *Actinidia08281.t1*,
	*Actinidia10240.t1*, *Actinidia13003.t1*, *Actinidia14109.t1*,
	*Actinidia20508.t1*, *Actinidia20994.t1*, *Actinidia24811.t1*,
	*Actinidia26254.t1*, *Actinidia26649.t1*, *Actinidia29458.t1*,
	*Actinidia36885.t1*, *Actinidia37473.t1*, *Actinidia39836.t2*

## Data Availability

Raw reads used in this work were deposited in NCBI Bio-Project under the accession numbers PRJNA796069, PRJNA514180, and PRJNA765913.

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
