# Peer review of "Genome-Wide Identification and Expression Analysis of Kiwifruit Leucine-Rich Repeat Receptor-Like Proteins Reveal Their Roles in Biotic and Abiotic Stress Responses"

_ijms, 2024, doi:10.3390/ijms25084497_

Round 1
Reviewer 1 Report
Comments and Suggestions for Authors
The author should address the following suggestion,
Line 20-21: Botanical name should be italic
Line 24: gene name should be italic and protein not italic, differentiate in whole abstract "LRR-RLP"
Arabidopsis should be italic in whole manuscript
Line 316: use approperiate term "king of Vitamin C"
in the conclusion seciton, remove repetation,
remove minor typos such as gene name
overall manuscript interesting and presenting very well,
Author Response
- Comment: Line 20-21: Botanical name should be italic.
Response:Thank you for the suggestion. We have revised the botanical names to ensure they are correctly formatted in italics.
Excerpt from the revised manuscript (Abstract):
We identified a total of 101, 164, and 105 LRR-RLPs in Actinidia Chinensis 'Hongyang', Actinidia eriantha 'Huate', and Actinidia chinensis 'Red5', respectively.
- Comment:Line 24: gene name should be italic and protein not italic, differentiate in whole abstract "LRR-RLP".
Response:Thank you for the suggestion. We have carefully reviewed the manuscript and made revisions to ensure proper formatting of gene names and protein names. We have also maintained consistency in differentiating the term "LRR-RLP" throughout the abstract.
Excerpt from the revised manuscript (Abstract):
Leucine-rich repeat receptor-like proteins (LRR-RLPs), a major group of receptor-like proteins in plants, have diverse functions in plant physiology, including growth, development, signal transduction, and stress responses. Despite their importance, the specific roles of kiwifruit LRR-RLPs in response to biotic and abiotic stresses remain poorly understood. In this study, we performed family identification, characterization, transcriptome data analysis and differential gene expression analysis of kiwifruit LRR-RLPs. We identified a total of 101, 164, and 105 LRR-RLPs in Actinidia Chinensis 'Hongyang', Actinidia eriantha 'Huate', and Actinidia chinensis 'Red5', respectively. Synteny analysis revealed that the expansion of kiwifruit LRR-RLPs was primarily attributed to segmental duplication events. Based on RNA-seq data from pathogen-infected kiwifruits, we identified specific LRR-RLP genes potentially involved in different stages of pathogen infection. Additionally, we observed the potential involvement of kiwifruit LRR-RLPs in abiotic stress responses, with upstream transcription factors possibly regulating their expression. Furthermore, protein interaction network analysis unveiled the participation of kiwifruit LRR-RLP in the regulatory network of abiotic stress responses. These findings highlight the crucial roles of LRR-RLPs in mediating both biotic and abiotic stress responses in kiwifruit, offering valuable insights for the breeding of stress-resistant kiwifruit varieties.
- Comment:Arabidopsis should be italic in whole manuscript.
Response: Thank you for the suggestion. We have ensured that “Arabidopsis” is italicized iconsistently throughout the entire manuscript. We appreciate your assistance in improving the quality of our manuscript.
- Comment: Line316:use approperiate term "king of Vitamin C".
Response: Thank you for your feedback and suggestion. We have replaced the sentence with a more appropriate term. We appreciate your suggestions to improve our manuscript.
Excerpt from the revised manuscript (Discussion):
Kiwifruit is an important economic crop.
- Comment:in the conclusion section, remove repetition.
Response: Thank you for the suggestion. We have revised the conclusion section to elimiate repetition. We appreciate your assistance in enhancing our manuscript.
Excerpt from the revised manuscript (Conclusions):
In conclusion, this study analyzed the kiwifruit LRR-RLP gene family members identification, physicochemical properties, phylogenetic relationships, chromosomal positions, and conducted their expression levels under biotic and abiotic stress. Segmental duplication emerged as the primary driver for the expansion of the LRR-RLP gene family in kiwifruit. The differentially expressed LRR-RLP genes in resistant and susceptible kiwifruits infected with Psa highlighted their involvement in biotic stress responses. We identified four LRR-RLP genes and related transcription factors that may be involved in pathogen stress. Additionally, the expression patterns of LRR-RLP genes under three abiotic stresses underscore their significance in abiotic stress tolerance. The protein interaction network elucidated the regulatory relationships between LRR-RLP members and associated proteins, shedding light on the complex network governing abiotic stress responses in kiwifruit. These findings provided valuable reference information for understanding protein interactions within the LRR-RLP gene family and contributed to the study of resistant kiwifruit varieties.
Reviewer 2 Report
Comments and Suggestions for Authors
Dear Editor
The manuscript “ Comprehensive Analysis of Leucine-rich Repeat Receptor-like Proteins Provides Insights into Their Roles in Biotic and Abiotic Stress Responses in Kiwifruit” in its current form cannot be proceeded with for further editorial stages.
Please find below my comments.
With best regards
Reviewer
Title
1. I propose to refine the topic according to the presenting results.
Abstract
2. The first sentence is too general and does not contribute substantive knowledge.
3. I propose to keep the abstract strictly structured:
(a) place the introduction addressed in a broad context,
(b) highlight the purpose of the study,
(c) describe briefly the main methods applied,
(d) summarise the main findings,
(e) and indicate the main conclusions.
Keywords
4. Please eliminate terms that appear in the title of the manuscript.
Introduction
l. 34-51.
5. Please explain which nutritional values, this is very important the latest scientific publications report on health-promoting phytotherapy.
6. What is the "...medicinal value...", please describe the effect (giving biological model: cell lines, animal studies, clinical studies).
7. Explain what is "...economic value" eliminate mental shortcuts.
8. Please state what economic losses "...leading to substantial eco-nomic losses in the industry on a worldwide scale...".
l. 52-70.
9. State which microorganisms, keeping a close link to the topic of the paper
"Although beneficial microorganisms contribute to plant growth and agricultural production, pathogenic ones can cause disease and economic losses".
10. Describe what the natural immune response of plants against potential pathogens is.
In plants, almost every cell is capable of eliciting an effective immune response by acting - system - zig-zag model.
l. 71-86.
11. The authors report effects in Arabidopsis thaliana, in plants of the genus Solanum and Oryza, please complete the mechanism of action with reference to the topic of the paper.
l. 87-93.
12 The authors describe "identified, revealing LRR-RLP members" in: A. thaliana, Brassica napus, Brassica juncea, Brassica rapa and Brassica nigra and plants of the genus Musa and Solanum, Oryza. Please complete the truncated reference to the manuscript theme.
l. 94-105
13 Please complete the scientific research thesis.
14. Formulate the specific aim of the work.
15. Move some information to the results subsection.
Results
l. 107-133
16. Please note the editing of the text (l. 116-129) analogous to the entire manuscript.
17. Figure 1. Insert consecutive letter designations for the individual graphs (a, b, c) and include in the description of the results in the text.
18. Figure 1. Insert X-axis designation on the graphs.
l. 134-235
19. Table 1. Please insert explanations of abbreviations below the table. It is assumed that the documentation of the study (table, graph) should be readable without looking elsewhere in the text, analogous to the whole manuscript.
20. Figure 4: The ordinal markings of the following graphs are too large, please standardise according to the guidelines for authors.
21. Figure 5. See notes above regarding figures.
l. 236-314.
22. Figure 8. The ordinal numbers of the following figures are too large, please normalise according to the authors' guidelines.
Discussion
l. 315-332.
23. Sentence was repeated in the introduction "Kiwifruit is known as the 'king of Vitamin C' due to its high nutritional, economic, and medicinal value."
24. Please provide your own research results first and then discuss them with the reports of other authors, but strictly referring to the topic of the paper.
25. Please change the review character of the text to a discussion, the remark applies to the entire discussion subsection.
l. 333-335.
26. No reference to results of other authors - no discussion.
l. 356-382.
27. Please complete the discussion.
Materials and Methods
l. 383-445
28. Please insert correct literature chat as required for Authors.
l. 446-456.
28. Please insert literature citation confirming methodological validity.
Conclusions
l. 457-470.
29. Please formulate specific conclusions resulting from each stage of the research.
30. Conclusions should respond to the aim of the work and the scientific research thesis.
References
31. l. 506, 507. please make the correction "K. Actinidia deliciosa (Kiwi fruit):".
32. The Latin names of the genus or species should be written with the perfective l. 542-544; 560, 561;
33. Please make a correction.
34. Please prepare a literature list according to the guidelines for authors, it is impossible to list all the notes.
Author Response
Response to Reviewer #2
Title:
1.Comment:I propose to refine the topic according to the presenting results.
Response: Thank you for your feedback. We have carefully reviewed the manuscript and have revised the title to better align with the presented results. We appreciate your assistance in improving the manuscript.
Excerpt from the revised manuscript (Title):
Genome-wide Identification and Expression Analysis of Kiwifruit Leucine-rich Repeat Receptor-like Proteins Reveals Their Roles in Biotic and Abiotic Stress Responses
Abstract:
2.Comment: The first sentence is too general and does not contribute substantive knowledge.
Response: Thank you for the valuable suggestion. After careful consideration, we acknowledge that this sentence is indeed dispensable. It has been removed in the revised manuscript.
3.Comment:I propose to keep the abstract strictly structured:
(a) place the introduction addressed in a broad context,
(b) highlight the purpose of the study,
(c) describe briefly the main methods applied,
(d) summarise the main findings,
(e) and indicate the main conclusions.
Response: Thank you for the valuable suggestion. We have readjusted the structure of the abstract section accordingly. We appreciate your assistance in enhancing the quality of our manuscript.
Excerpt from the revised manuscript (Abstract):
Leucine-rich repeat receptor-like proteins (LRR-RLPs), a major group of receptor-like proteins in plants, have diverse functions in plant physiology, including growth, development, signal transduction, and stress responses. Despite their importance, the specific roles of kiwifruit LRR-RLPs in response to biotic and abiotic stresses remain poorly understood. In this study, we performed family identification, characterization, transcriptome data analysis and differential gene expression analysis of kiwifruit LRR-RLPs. We identified a total of 101, 164, and 105 LRR-RLPs in Actinidia Chinensis 'Hongyang', Actinidia eriantha 'Huate', and Actinidia chinensis 'Red5', respectively. Synteny analysis revealed that the expansion of kiwifruit LRR-RLPs was primarily attributed to segmental duplication events. Based on RNA-seq data from pathogen-infected kiwifruits, we identified specific LRR-RLP genes potentially involved in different stages of pathogen infection. Additionally, we observed the potential involvement of kiwifruit LRR-RLPs in abiotic stress responses, with upstream transcription factors possibly regulating their expression. Furthermore, protein interaction network analysis unveiled the participation of kiwifruit LRR-RLP in the regulatory network of abiotic stress responses. These findings highlight the crucial roles of LRR-RLPs in mediating both biotic and abiotic stress responses in kiwifruit, offering valuable insights for the breeding of stress-resistant kiwifruit varieties.
Keywords:
4.Comment:Please eliminate terms that appear in the title of the manuscript.
Response:Thank you for the good suggestion. We have removed terms that appear in the title from the keywords list and updated the manuscript accordingly.
Excerpt from the revised manuscript (Keywords):
Keywords: kiwifruit; LRR-RLP; stress; differentially expressed gene; plant immunity
Introduction:
5.Comment: Please explain which nutritional values, this is very important the latest scientific publications report on health-promoting phytotherapy.
Response: Thank you for your attention. Kiwifruit is known fro its high content of vitamin C, minerals, dietary fiber, and other beneficial metabolites (Huang et al, 2013). These nutritional aspects highlight the health-promoting properties of kiwifruit.
Reference:
Huang, S.; Ding, J.; Deng, D.; Tang, W.; Sun, H.; Liu, D.; Zhang, L.; Niu, X.; Zhang, X.; Meng, M.; et al. Draft genome of the kiwifruit Actinidia chinensis. Nat. Commun. 2013, 4, 2640.
6.Comment:What is the "...medicinal value...", please describe the effect (giving biological model: cell lines, animal studies, clinical studies).
Response: Thank you for your attention. Satpal(2021) described that “Kiwifruit and its components exhibit various pharmacological properties, including antidiabetic, anti-tumor, anti-inflammatory, anti-ulcer, antioxidant activity, hypoglycemic, hypolipidemic effects, and more”. We have supplemented the reference in the revised manuscript.
Reference:
Satpal D, Kaur J, Bhadariya V, et al. Actinidia deliciosa(Kiwi fruit): A comprehensive review on the nutritional composition, health benefits, traditional utilization, and commercialization[J]. Journal of Food Processing and Preservation, 2021, 45(6):e15588.
7.Comment:Introduction:Explain what is "...economic value" eliminate mental shortcuts.
Response:Thank you for your attention. Kiwifruit has established a stable position in the fresh fruit market, and it is utilized in the production of products including drinks, confectionery, yogurts, soaps, shampoos, and other cosmetic products(Satpal et al, 2021). The article discusses the stability of kiwifruit in the market and the diversification of kiwifruit products. These products contribute to the overall economic value of kiwifruit.
Reference:
Satpal D, Kaur J, Bhadariya V, et al. Actinidia deliciosa(Kiwi fruit): A comprehensive review on the nutritional composition, health benefits, traditional utilization, and commercialization[J]. Journal of Food Processing and Preservation, 2021, 45(6):e15588.
8.Comment:Please state what economic losses "...leading to substantial eco-nomic losses in the industry on a worldwide scale...".
Response: Thank you for your attention. Since the economically devastating Psa outbreak in Japan in the 1980s, the disease began to spread, causing severe economic losses to the kiwifruit industry in many countries (Cameron et al, 2014). Psa, the pathogen responsible for kiwifruit bacterial canker, can lead to the destruction of kiwifruit vines or orchards, resulting in significant economic losses (Kim et al, 2017).
Reference:
Cameron, A.; Sarojini, V. Pseudomonas syringae pv. actinidiae: chemical control, resistance mechanisms and possible alterna-tives. Plant Pathol, 2014, 63: 1-11.
Kim GH, Jung JS, Koh YJ. Occurrence and Epidemics of Bacterial Canker of Kiwifruit in Korea. Plant Pathol J, 2017, 33(4):351-361.
9.Comment:State which microorganisms, keeping a close link to the topic of the paper" Although beneficial microorganisms contribute to plant growth and agricultural production, pathogenic ones can cause disease and economic losses".
Response: Thank you for your comment on our study. Among the microorganisms surrounding the rhizosphere, plant growth-promoting rhizobacteria (PGPR) are extensively studied for their beneficial effects (Jeon et al, 2022). Additionally, it's worth noting that Psa is the pathogen responsible for kiwifruit bacterial canker (Kim et al, 2017), which directly related to the focus of our study.
Reference:
Jeon, JS., Rybka, D., Carreno-Quintero, N., et al. (2022) Metabolic signatures of rhizobacteria-induced plant growth promotion. Plant, Cell & Environment, 2022, 45: 3086–3099.
Kim GH, Jung JS, Koh YJ. Occurrence and Epidemics of Bacterial Canker of Kiwifruit in Korea. Plant Pathol J, 2017,33(4):351-361.
10.Comment:Describe what the natural immune response of plants against potential pathogens is.In plants, almost every cell is capable of eliciting an effective immune response by acting - system - zig-zag model
Response:Thank you for your comment on our study. Plants lack a circulating immune system and rely on the ability of each cell to deploy innate immune responses against potential pathogens(Couto &Zipfel, 2016). The dynamic co-evolutionary relationship between plants and pathogens can be summarized by a four-stage ‘zigzag’ model(Jones &Dangl, 2006). In the first stage, pattern recognition receptors(PRRs) recognizes specific molecular patterns and activates pattern-triggered immunity (PTI) to inhibit pathogen replication and disease development. In the second stage, pathogens evade or inhibit PRR recognition by secreting effector factors into host cells, leading to effector-triggered susceptibility (ETS). In the third stage, plants have evolved nucleotide-binding and leucine-rich repeat receptors (NLRs) to specifically recognize effectors, inducing effector-triggered immunity (ETI) to prevent further pathogen infection. In the fourth stage, pathogens may circumvent plant ETI by evolving existing effectors or acquiring new effectors, promoting the plant to evolve new NLR proteins to reactivate ETI (Jones &Dangl, 2006). This intricate interaction among PTI, ETS, and ETI constitutes the ‘zigzag’ model(Jones &Dangl, 2006).
Reference:
Couto, D., & Zipfel, C. Regulation of pattern recognition receptor signalling in plants. Nature Reviews Immunology, 2016,16(9): 537–552.
Jones, JD., & Dangl, JL. The plant immune system. Nature, 2006,444(7117): 323–329.
11.Comment:The authors report effects in Arabidopsis thaliana, in plants of the genus Solanum and Oryza, please complete the mechanism of action with reference to the topic of the paper
Response: Thank you for your feedback and suggestion. In this paragraph, we mainly explain that LRR-RLPs play crucial roles in growth and development, signal transduction, and stress response, indicating their research significance in plants.
12.Comment:The authors describe "identified, revealing LRR-RLP members" in: A. thaliana, Brassica napus, Brassica juncea, Brassica rapa and Brassica nigra and plants of the genus Musa and Solanum, Oryza. Please complete the truncated reference to the manuscript theme.
Response: Thank you for your valuable suggestion. In this paragraph, we mainly illustrate that LRR-RLPs have been studied in multiple species. The species involved have also been cited in the relevant references, highlighting their significance in understanding the theme of the manuscript.
13.Comment:Please complete the scientific research thesis.
Response: Thank you for your feedback and suggestion. We apologize for any lack of precision in our previous presentation and have improved this part of the research content accordingly.
Excerpt from the revised manuscript (Introduction):
Kiwifruit is vulnerable to both biotic and abiotic stresses. LRR-RLP is crucial in plant growth and development, signal transduction, and stress responses. However, its study in kiwifruit has received limited attention. This study delves into the analysis of the LRR-RLP genes in kiwifruit, encompassing the identification of gene family members, phylogenetic analysis, chromosomal localization, synteny analysis, transcriptome data analysis, upstream transcription factor analysis, and construction of a protein interaction network. Our findings contribute to a deeper understanding of the defense mechanisms employed by kiwifruit, holding crucial reference value for the genetic enhancement of kiwifruit varieties. Moreover, this study provides valuable insights for the development of resistant kiwifruit varieties.
14.Comment: Formulate the specific aim of the work.
Response: Thank you for the suggestion. We apologize for the lack of clarity in stating the work goal in our previous version. We have rewritten it to ensure it is clear and updated the manuscript accordingly.
Excerpt from the revised manuscript (Introduction):
Kiwifruit is vulnerable to both biotic and abiotic stresses. LRR-RLP is crucial in plant growth and development, signal transduction, and stress responses. However, its study in kiwifruit has received limited attention. This study delves into the analysis of the LRR-RLP genes in kiwifruit, encompassing the identification of gene family members, phylogenetic analysis, chromosomal localization, synteny analysis, transcriptome data analysis, upstream transcription factor analysis, and construction of a protein interaction network. Our findings contribute to a deeper understanding of the defense mechanisms employed by kiwifruit, holding crucial reference value for the genetic enhancement of kiwifruit varieties. Moreover, this study provides valuable insights for the development of resistant kiwifruit varieties.
15.Comment:Move some information to the results subsection.
Response: Thank you for your feedback. We have revised the manuscript according to your suggestion.
Excerpt from the revised manuscript (Introduction):
Kiwifruit is vulnerable to both biotic and abiotic stresses. LRR-RLP is crucial in plant growth and development, signal transduction, and stress responses. However, its study in kiwifruit has received limited attention. This study delves into the analysis of the LRR-RLP genes in kiwifruit, encompassing the identification of gene family members, phylogenetic analysis, chromosomal localization, synteny analysis, transcriptome data analysis, upstream transcription factor analysis, and construction of a protein interaction network. Our findings contribute to a deeper understanding of the defense mechanisms employed by kiwifruit, holding crucial reference value for the genetic enhancement of kiwifruit varieties. Moreover, this study provides valuable insights for the development of resistant kiwifruit varieties.
Results:
16.Comment:Please note the editing of the text (l. 116-129) analogous to the entire manuscript.
Response: Thank you for the suggestion. This paragraph describes physicochemical properties of three kinds of kiwifruit LRR-RLPs, and the sentence structure is somewhat similar. We have readjusted the sentence structure and updated the manuscript.
Excerpt from the revised manuscript (Results2.1):
We also conducted an analysis of the physicochemical properties of LRR-RLP members in these three kiwifruit species (Figure 1). Detailed physicochemical properties of HyLRR-RLP, HtLRR-RLPs and DhLRR-RLPs are provided in Table S1, S2, and S3, respectively. The amino acid sequence lengths of HyLRR-RLPs ranged from 117 to 2136, with the protein molecular weights (MW) ranging from 12.60 to 234.25 kDa, and isoelectric points(PI) ranging between 4.45 and 10.3. For HtLRR-RLPs, amino acid sequence lengths ranged from 74 to 1260, the protein MW were from 8.29 to 138.89 kDa, and PI ranged between 4.3 and 12.11. The amino acid sequences of DhLRR-RLPs ranged from 77 to 1180, and the protein MW varied from 8.90 to 129.82 kDa, with PI between 4.19 and 9.51.
17.Comment: Figure 1. Insert consecutive letter designations for the individual graphs (a, b, c) and include in the description of the results in the text.
Response: Thank you for your attention. We have revised Figure 1 according to your suggestions,. We appreciate your assistance in improving the manuscript.
Excerpt from the revised manuscript (Figure 1.):
18.Comment: Figure 1. Insert X-axis designation on the graphs.
Response: Thank you for the suggestion. We have revised Figure 1 to include X-axis designation on the graphs.
Excerpt from the revised manuscript (Figure 1.):
19.Comment: Table 1. Please insert explanations of abbreviations below the table. It is assumed that the documentation of the study (table, graph) should be readable without looking elsewhere in the text, analogous to the whole manuscript.
Response: Thank you for the suggestion. We have revised Table 1 according to your suggestions to include explanations of abbreviations below the table, ensuring consistency with the rest of the manuscript.
Excerpt from the revised manuscript (Table 1. Notes):
Notes: HyLRR-RLPs represent the LRR-RLPs of Actinidia Chinensis 'Hongyang', HtLRR-RLPs represent the LRR-RLPs of Actinidia eriantha 'Huate', DhLRR-RLPs reprenst the LRR-RLPs of Actinidia chinensis 'Red5'.
20.Comment: Figure 4: The ordinal markings of the following graphs are too large, please standardise according to the guidelines for authors.
Response: Thank you for the suggestion. We have adjusted the size of the ordinal markings in Figure 4 to adhere to the guidelines. We appreciate your suggestions to enhance the visual presentation of our Figure.
Excerpt from the revised manuscript (Figure 4):
21.Comment:Figure 5. See notes above regarding figures.
Response: Thank you for the suggestion. We have revised Figure 5 accordingly.
Excerpt from the revised manuscript (Figure 5):
22.Comment:Figure 8. The ordinal numbers of the following figures are too large, please normalise according to the authors' guidelines.
Response: Thank you for the suggestion. We have adjusted the size of the ordinal markings in Figure 8 to align with the guidelines. We appreciate your suggestions to enhance the visual presentation of our figure.
Excerpt from the revised manuscript (Figure 8):
Discussion:
23.Comment: Sentence was repeated in the introduction "Kiwifruit is known as the 'king of Vitamin C' due to its high nutritional, economic, and medicinal value."
Response: Thank you for your good suggestion. We have removed the repetitive sentence and replaced it with another one. We appreciate your help in improving the flow of the manuscript.
Excerpt from the revised manuscript (Discussion):
Kiwifruit is an important economic crop.
24.Comment: Please provide your own research results first and then discuss them with the reports of other authors, but strictly referring to the topic of the paper.
Response: Thank you for the suggestion. We have restructured the discussion section accordingly, presenting our own research results first before discussing them in conjunction with reports from other authors. We appreciate your suggestions to improve the manuscript.
25.Comment: Please change the review character of the text to a discussion, the remark applies to the entire discussion subsection.
Response: Thank you for the suggestion. We have revised this part of the manuscript according to your suggestion, changing he review character to a discussion throughout the discussion subsection.
26.Comment:No reference to results of other authors - no discussion.
Response: Thank you for your suggestion. We have incorporated relevant findings from other studies with references in our discussion section to provide a comprehensive analysis of the topic. Your feedback is valuable, and we appreciate the opportunity to enhance the quality of our work.
27.Comment:Please complete the discussion.
Response: Thank you for the suggestion. We have expanded the content in the discussion section. We appreciate your suggestions to improve the manuscript.
Excerpt from the revised manuscript (Discussion):
We speculate that LRR-RLP and its related upstream transcription factors play an important role in kiwifruit response to biotic and abiotic stresses.
Materials and Methods:
28.Comment: Please insert correct literature chat as required for Authors.Please insert literature citation confirming methodological validity.
Response: Thank you for your valuable feedback on the manuscript. We appreciate your attention to detail and your suggestions for improvement. We have reviewed the literature review section to ensure that it comprehensively covers all relevant studies and theories essential for this work. We have cited new references to ensure the reliability of the methodology.
Excerpt from the revised manuscript (References):
Szklarczyk, D.; Kirsch, R.; Koutrouli, M.; Nastou, K.; Mehryary, F.; Hachilif, R.; Gable, A. L.; Fang, T.; Doncheva, N. T.; Pyysalo, S.; et al. The STRING database in 2023: protein-protein association networks and functional enrichment analyses for any sequenced genome of interest. Nucleic Acids Res.2023, 51, D638–D646.
Conclusions:
29.Comment:Please formulate specific conclusions resulting from each stage of the research.
Response: Thank you for the suggestion. We acknowledge the importance of formulating specific conclusions for each stage of the research to enhance the clarity and coherence of the manuscript. In response to your comment, we have revisited the manuscript and refined the conclusions for each research stage.
Excerpt from the revised manuscript (Conclusions):
In conclusion, this study analyzed the kiwifruit LRR-RLP gene family members' identification, physicochemical properties, phylogenetic relationships, chromosomal positions, and conducted their expression levels under biotic and abiotic stress. Segmental duplication emerged as the primary driver for the expansion of the LRR-RLP gene family in kiwifruit. The differentially expressed LRR-RLP genes in resistant and susceptible kiwifruits infected with Psa highlighted their involvement in biotic stress responses. We identified four LRR-RLP genes and related transcription factors that may be involved in pathogen stress. Additionally, the expression patterns of LRR-RLP genes under three abiotic stresses underscore their significance in abiotic stress tolerance. The protein interaction network elucidated the regulatory relationships between LRR-RLP members and associated proteins, shedding light on the complex network governing abiotic stress responses in kiwifruit. These findings provided valuable reference information for understanding protein interactions within the LRR-RLP gene family and contributed to the study of resistant kiwifruit varieties.
30.Comment:Conclusions should respond to the aim of the work and the scientific research thesis.
Response: Thank you for the suggestion. We have revised the conclusion according to your suggestion to ensure it aligns with the aim of the work and the scientific research thesis. We appreciate your assistance in improving the manuscript.
Excerpt from the revised manuscript (Conclusions):
In conclusion, this study analyzed the kiwifruit LRR-RLP gene family members' identification, physicochemical properties, phylogenetic relationships, chromosomal positions, and conducted their expression levels under biotic and abiotic stress. Segmental duplication emerged as the primary driver for the expansion of the LRR-RLP gene family in kiwifruit. The differentially expressed LRR-RLP genes in resistant and susceptible kiwifruits infected with Psa highlighted their involvement in biotic stress responses. We identified four LRR-RLP genes and related transcription factors that may be involved in pathogen stress. Additionally, the expression patterns of LRR-RLP genes under three abiotic stresses underscore their significance in abiotic stress tolerance. The protein interaction network elucidated the regulatory relationships between LRR-RLP members and associated proteins, shedding light on the complex network governing abiotic stress responses in kiwifruit. These findings provided valuable reference information for understanding protein interactions within the LRR-RLP gene family and contributed to the study of resistant kiwifruit varieties.
References:
31.Comment:l. 506, 507. please make the correction "K. Actinidia deliciosa (Kiwi fruit):"
Response:Thank you for bringing this to our attention. The original reference“Actinidia deliciosa (Kiwi fruit)”is written in the specific format with Actinidia deliciosa italicized and ‘Kiwi fruit’ separated by a space. We appreciate your valuable suggestions to improve the manucript.
32.Comment: The Latin names of the genus or species should be written with the perfective l. 542-544; 560, 561;
Response: Thank you for the suggestion. We have revised the manuscript accordingly to ensure that the names are presented accurately.
33.Comment:Please make a correction.
Response: Thank you for the suggestion. We have thoroughly reviewed the entire manuscript and corrected any inaccuracies. We appreciate your assistance in improving the manuscript.
34.Comment:Please prepare a literature list according to the guidelines for authors, it is impossible to list all the notes.
Response: Thank you for your valuable feedback. We acknowledge your suggestion and have prepared a literature list in accordance with the guidelines for authors. We have included the primary references cited within the text while adhering to the journal's formatting requirements.

Reviewer 3 Report
Comments and Suggestions for Authors
General comments
The manuscript includes a complex examination of the LRR-RLP family in kiwifruit based on genome and transcriptome analysis. A number of LRR-RLP genes in the genomes of three kiwifruit species are identified. Their physicochemical properties, evolutionary relationships and collinearity relationships are evaluated. Evidence for the role of LRR-RLPs in mediating the response to biotic and abiotic stresses of the plants is presented.
The Abstract pointed the main results of the study.
The Introduction is informative and is focused on the topic of the research.
The Results section is well structured. A sufficient number of figures, tables and supplementary materials are included.
The Discussion part summarizes the most important results and benefits of the study.
The Material and Methods are described in detail.
References are up to date.
Specific comments
Some small technical inaccuracies, comments and suggestions to the authors concerning expression profiles of LRR-RLPs in response to P. syringae infection are pointed below.
Page 4, Rows 138-140
There are two sentences with the same meaning.
Section 2.5.
LRR-RLP genes Actinidia39875.t1 and Actinidia06882.t1 were determined as simultaneously differentially expressed in response to P. syringae infection in both susceptible Hy and resistant Ht kiwifruit (rows 204-207). After that detailed information about expression changes of Actinidia39875.t1 and other three genes are presented on Figure 5, but not for Actinidia06882.t1. This seems like a mismatch. Are the expression levels of the two genes similar?
Moreover, the expression change of selected genes on Figure 5 should be represented more accurately because the description does not fully match the data in the graphs (rows 317-235). There are differences in gene expression in Hy compared to 0h not only for Actinidia39875.t1, but also for other three genes. The expression of Actinidia35026.t1 is determined as “stable” and “relatively constant” for Actinidia12020.t1 and Actinidia34674.t1 in Hy, however the difference looks statistically significant between 0h and the other time points. In addition, these two genes are differentially up-regulated not only at 48h and 96h time points in Ht, but also at earlier hours of infection, which is not pointed by the authors.
Apart from the role of selected genes in the early or late stages of the infection process, a connection between expression models and resistance/susceptibility of Ht and Hy, respectively, to the pathogen, should be indicated too.
Author Response
Response to Reviewer #3
1.Comment:The manuscript includes a complex examination of the LRR-RLP family in kiwifruit based on genome and transcriptome analysis. A number of LRR-RLP genes in the genomes of three kiwifruit species are identified. Their physicochemical properties, evolutionary relationships and collinearity relationships are evaluated. Evidence for the role of LRR-RLPs in mediating the response to biotic and abiotic stresses of the plants is presented.
The Abstract pointed the main results of the study.
The Introduction is informative and is focused on the topic of the research.
The Results section is well structured. A sufficient number of figures, tables and supplementary materials are included.
The Discussion part summarizes the most important results and benefits of the study.
The Material and Methods are described in4rrrrrrrr detail.
References are up to date.
Response: Thanks for the positive feedback and valuable suggestions. We have checked through the manuscript and corrected some typos and grammar errors. Additionally, we have revised sentences to enhance readability in the revised manuscript. We appreciate your careful review and assistance in maintaining the accuracy and integrity of the work.
2.Comment:Page 4, Rows 138-140. There are two sentences with the same meaning.
Response: Thank you for bringing this to our attention. We apologize for the redundancy, which was due to our carelessness. We have removed the duplicate sentence to improve clarity and conciseness. Your assistance in improving the manuscript is greatly appreciated.
3.Comment:LRR-RLP genes Actinidia39875.t1 and Actinidia06882.t1 were determined as simultaneously differentially expressed in response to P. syringae infection in both susceptible Hy and resistant Ht kiwifruit (rows 204-207). After that detailed information about expression changes of Actinidia39875.t1 and other three genes are presented on Figure 5, but not for Actinidia06882.t1. This seems like a mismatch. Are the expression levels of the two genes similar?
Response: Thank you for your detailed analysis. We apologize for the oversight in not providing specific information about the expression changes of Actinidia06882.t1 in Figure 5. We have examined the expression levels of both Actinidia39875.t1 and Actinidia06882.t1 in response to P. syringae infection in both susceptible Hy and resistant Ht kiwifruit. Upon comparison, we found that the expression levels of Actinidia39875.t1 and Actinidia06882.t1 are indeed different. Specifically, the expression trend of Actinidia39875.t1 varies between susceptible and resistant kiwifruit, while the expression trend of Actinidia06882.t1 remains consistent across both types of kiwifruit. We believe that Actinidia39875.t1 may play a different role in susceptible and resistant kiwifruit, whereas Actinidia06882.t1 may have a consistent role in both. We have made the necessary adjustments to clarify this distinction in our manuscript. Thank you for bringing this to our attention and helping us improve the accuracy of our work.
4.Comment:Moreover, the expression change of selected genes on Figure 5 should be represented more accurately because the description does not fully match the data in the graphs (rows 317-235). There are differences in gene expression in Hy compared to 0h not only for Actinidia39875.t1, but also for other three genes. The expression of Actinidia35026.t1 is determined as “stable” and “relatively constant” for Actinidia12020.t1 and Actinidia34674.t1 in Hy, however the difference looks statistically significant between 0h and the other time points. In addition, these two genes are differentially up-regulated not only at 48h and 96h time points in Ht, but also at earlier hours of infection, which is not pointed by the authors.
Response: Thank you for your detailed feedback. We have carefully reviewed the expression changes of the selected genes in Figure 5. Upon examination, we acknowledge that there are discrepancies between the descriptions and the data presented in the graphs. Compared with 0h, Actinidia39875.t1 was differentially expressed in Hy and Ht at all time nodes, but the expression of Actinidia39875.t1 was higher in Ht than in Hy. The expression of Actinidia39875.t1 showed a gradually increasing trend in Hy, but it reached the maximum at 12h and then gradually decreased in Ht. Actinidia35026.t1, Actinidia12020.t1 and Actinidia34764.t1 were not differentially expressed in Hy, but all three were differentially expressed at 48h and 96h in Ht. The expression of Actinidia35026.t1 in Ht decreased gradually. The expression levels of Actinidia12020.t1 and Actinidia34764.t1 showed a gradually increasing trend in Ht. In summary, we speculated that Actinidia39875.t1 and Actinidia35026.t1 play a role in the early stage of Psa infection, while Actinidia12020.t1 and Actinidia34674.t1 play a role in the late stage of Psa infection. We appreciate your thorough review and your efforts to improve the accuracy of the work.
5.Comment: Apart from the role of selected genes in the early or late stages of the infection process, a connection between expression models and resistance/susceptibility of Ht and Hy, respectively, to the pathogen, should be indicated too.
Response: Thank you for your insightful comment. We agree that it is important to consider the connection between gene expression patterns and the resistance/susceptibility of kiwifruit varieties to the pathogen. Based on our analysis, we found that certain LRR-RLP genes were differentially expressed in resistant kiwifruit(Ht) compared to susceptible kiwifruit(Hy). This suggests a potential association between the expression levels of these genes and the resistance to Psa in kiwifruit. We speculated that the higher expression level of these LRR-RLPs may confer greater resistance to Psa infection. We have included this interpretation in our manuscript to provide a comprehensive understanding of the relationship between gene expression and resistance/susceptibility to the pathogen. We appreciate your valuable input in enhancing the quality of the manuscript.